# Rolling Ball Optimizer: Learning by ironing out loss landscape wrinkles

## Abstract

Training large neural networks (NNs) requires optimizing high-dimensional data-dependent loss functions. The optimization landscape of these functions is often highly complex and textured, even fractal-like, with many spurious (sometimes sharp) local minima, ill-conditioned valleys, degenerate points, and saddle points. Complicating things further is the fact that these landscape characteristics are a function of the training data, meaning that noise in the training data can propagate forward and give rise to unrepresentative small-scale geometry. This poses a difficulty for gradient-based optimization methods, which rely on local geometry to compute their updates and are, therefore, vulnerable to being derailed by noisy data. In practice, this translates to a strong dependence of the optimization dynamics on the noise in the data, i.e., poor generalization performance. To remediate this problem, we propose a new optimization procedure: Rolling Ball Optimizer (RBO), that breaks this spatial locality by explicitly incorporating information from a larger region of the loss landscape in its updates. We achieve this by simulating the motion of a rigid sphere of finite radius $\rho > 0$ rolling on the loss landscape, a straightforward generalization of Gradient Descent (GD) that simplifies into it in the *infinitesimal* limit ($\rho \to 0$). The radius serves as a hyperparameter that determines the scale at which RBO "sees" the loss landscape, allowing control over the granularity of its interaction therewith. We are motivated in this work by the intuition that the large-scale geometry of the loss landscape is less data-specific than its fine-grained structure, and that it is easier to optimize. We support this intuition by proving that our algorithm has a smoothing effect on the loss function. Evaluation against SGD, SAM, and Entropy-SGD, on MNIST and CIFAR-10/100 demonstrates promising results in terms of convergence speed, training accuracy, and generalization performance. Our code is available at
https://anonymous.4open.science/r/rolling-ball-DC88/

## 1 Introduction

Training Deep Learning (DL) models fundamentally boils down to minimizing a data-dependent training loss over a hypothesis class, often parameterized by a parameter vector $\theta \in \mathbb{R}^d$, where $d$ is the dimensionality of the parameter space. In practice, $d$ is often extremely large, with the largest models now exceeding $d = 10^{13}$ parameters (Chenna, 2023). The differentiability[1] of the most common loss functions in DL, along with the ability to efficiently compute their gradients through automatic differentiation, makes them a natural target for gradient-based iterative optimization (Abdulkadirov et al., 2023).

However, a deeper examination of the training objective reveals fundamental differences from the typical setting where Gradient Descent (GD) is applied. On one hand, the data-dependence of the loss function makes it, and by extension, its gradients, noisy. Therefore, minimizing it directly via GD need not translate to good average performance on unobserved data. On the other hand, the loss landscapes of most deep neural networks (DNNs) have complex geometries, which are often highly non-convex, with many spurious local minima, some of which are sharp. This negates the classical global convergence guarantees of GD, resulting in convergence to a local minimum for small learning rates, and divergence for large ones.

---

[1]For many popular activation functions like $\mathrm{ReLU}$, we only have piecewise differentiability.

Multiple variants of GD have been proposed to address these issues. The most popular approaches include stochastic gradient methods like Stochastic Gradient Descent (SGD), which leverage noisy gradient estimates to escape local minima, momentum methods like Heavyball Ball (HB) (Polyak, 1964) and Nesterov Accelerated Gradient (NAG) (Nesterov, 1983), which use past gradients to determine update directions, and adaptive gradient methods like Root Mean Square Propagation (RMSProp) (Hinton, 2012) and Adaptive Gradient (AdaGrad) (Duchi et al., 2011) which use past gradients to dynamically adjust learning rates. While these optimizers, particularly, adaptive gradient methods do achieve their stated goal of improving optimization dynamics, they often do so at the expense of worse generalization performance, on which, they are often outperformed by SGD (Wilson et al., 2017; Zhou et al., 2020).

A key similarity between all the optimizers discussed thus far, which, we argue, is essential to their limitations, is their *"point-like"* nature. By this, we mean the fact that their dynamics resemble[2] those of a point-particle moving on the loss landscape under local forces (Kovachki & Stuart, 2021; Tanaka & Kunin, 2021; Heredia, 2024). In the sequel, we refer to this property as *spatial locality*. Due to spatial locality, the dynamics of point-like optimizers are sensitive to arbitrarily small perturbations in the loss function, including those that are due to the noise in training data, which goes a long way towards explaining why they struggle to generalize. Moreover, point-like optimizers are unable to capture the global geometry of the loss landscape. They can fall into arbitrarily sharp and/or ill-conditioned valleys (Keskar et al., 2022; Hochreiter & Schmidhuber, 1997; Grosse, 2021), get trapped by a saddle point of high index (Dauphin et al., 2014; Choromanska et al., 2015), or even behave chaotically due to the fractal-like geometry of the loss landscape (Lowell, 2022; Li et al., 2018; Chen et al., 2022; Herrmann et al., 2022). Combined, these two effects make point-like optimizers simultaneously sensitive to the microscopic structure of the loss landscape, which is likely due to noise, and oblivious to its macroscopic geometry, which is likely representative of genuine structure in the data. From this perspective, their generalization struggles are unsurprising.

More recently, a new class of optimizers has been proposed, which abandons spatial locality in favor of a more global view of the loss landscape. Most notable amongst these optimizers are Entropy-SGD (Chaudhari et al., 2017) and Sharpness-Aware Minimization (SAM) (Foret et al., 2020). Both optimizers are specifically designed to avoid sharp minima, with the aim of improving generalization. In the case of Entropy-SGD, this is done by optimizing over a smoothed version of the loss function via Langevin dynamics. As for SAM, it explicitly adds a sharpness penalty to its loss function, and minimizes an approximate version thereof directly. Both of these algorithms, but SAM in particular, compare favorably against spatially local optimizers on a multitude of benchmarks. However, their complete focus on avoiding sharp minima to the neglect of all other geometric properties of the loss landscape is, we argue, a limitation to their power.

In this paper, we propose a new spatially non-local optimizer, which considers the entire geometry in its neighborhood instead of just sharpness. The remedy we offer for the ills of point-like dynamics is to replace the point particle with a ball of finite radius $\rho > 0$ rolling *over* the loss landscape. The dynamics of such a ball respond to loss landscape features of size proportional to $\rho$, meaning that, by tuning the radius up or down, we can set the scale at which the optimizer interacts with the loss landscape. In particular, adding noise on a scale negligible compared to $\rho$ leaves the ball's trajectory approximately unchanged compared to the noiseless loss landscape. Additionally, this optimizer, which we call the Rolling Ball Optimizer (RBO), is immune to falling into sharp minima, and ill-conditioned valleys if they are too narrow to fit the ball, since it is not allowed to "phase through" the loss landscape. Finally, it has a smoothing effect on the loss landscape, making it stable with much larger learning rates than point-like optimizers. In fact, most of RBO's aforementioned properties can be seen as consequences of its smoothing effect.

## 2   ROLLING BALL OPTIMIZER (RBO)

We place ourselves in the typical DL setting, described in Section 1: a positive (sufficiently) differentiable loss function $f : \mathbb{R}^d \to \mathbb{R}$, is given, and its graph $\Gamma = \left\{ (\theta, f(\theta)) \mid \theta \in \mathbb{R}^d \right\} \subset \mathbb{R}^{d+1}$, is referred to as the loss landscape, loss manifold, or loss surface. RBO models the motion of a sphere moving on $\Gamma$, maintaining tangency at all times, never leaving or crossing it. Unlike point-like opti-

---

[2]And indeed, reduce to, in the so-called continuous-time limit.

mizers, which follow unconstrained trajectories in parameter space, the previous condition imposes on RBO a fundamental invariant. Namely, for all times $t \geq 0$, the distance between the center of the ball $c_t$ and the loss landscape $\Gamma$ must equal a constant: the radius $\rho$ of the ball. In symbols:

$$\forall t \geq 0, \quad d(c_t, \Gamma) := \inf_{p \in \Gamma} \|p - c_t\| = \rho. \tag{1}$$

Aside from the above, RBO behaves identically to SGD. It is therefore this constraint that breaks spatial locality. In Section 3, we will see that Equation (1) is equivalent to $c_t$ being confined to an *offset manifold* of $\Gamma$.

Enforcing Equation (1) during optimization is a non-trivial task. Collision detection and handling is already a hard problem in computer graphics and computational physics (Bender et al., 2015), and the high dimensionality of $\mathbb{R}^d$ certainly makes it no easier. The solution we adopt is based on "Marble Marcher"[3], an open source game by Code Parade that involves ball motion on a fractal surface. In concrete terms, our algorithm iteratively alternates between two steps. First, a descent step, similar to the GD update rule:

$$\widetilde{c}_{t+1} = c_t - \eta \tau(p_t), \tag{2}$$

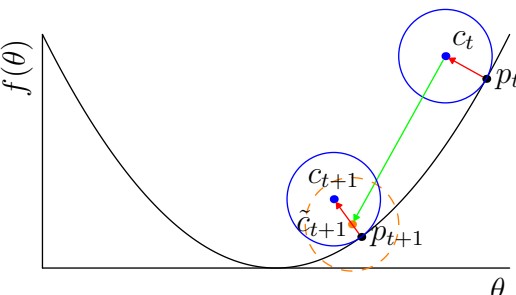

Figure 1: One update step of the RBO.

where $p_t$ is the contact point[4] of the ball with $\Gamma$ at time step $t$, $\eta$ is the learning rate, and for a point $p = (\theta, f(\theta)) \in \Gamma$, $\tau(p)$ and $\nu(p)$ are given respectively by

$$\tau(p) = \begin{bmatrix} \nabla f(\theta) \\ \|\nabla f(\theta)\|^2 \end{bmatrix}, \tag{3}$$

and

$$\nu(p) = \frac{1}{\sqrt{1 + \|\nabla f(\theta)\|^2}} \begin{bmatrix} -\nabla f(\theta) \\ 1 \end{bmatrix}. \tag{4}$$

This step almost always breaks Equation (1) (see dashed circle in Figure 1). Second, to restore the distance constraint, a *constraint projection* must be applied to Equation (2). Following the Marble Marcher example, this is achieved by finding the closest point on $\Gamma$ to the candidate center $\widetilde{c}_{t+1}$, and making it the new contact point $p_{t+1}$ as follows:

$$p_{t+1} = \underset{p \in \Gamma}{\operatorname{argmin}} \|p - \widetilde{c}_{t+1}\|^2. \tag{5}$$

The new center is then given by $c_{t+1} = p_{t+1} + \rho\nu(p_{t+1})$, where $\nu(p)$ is the upwards pointing unit normal to $\Gamma$ at $p$ (see Figure 1). These two steps together are then repeated until a stopping criterion is met, giving rise to Algorithm 1, a Projected Gradient Descent (PGD) method that approximately satisfies Equation (1).

As previously noted, the constraint projection step is what distinguishes RBO from GD. As a result, it must be responsible for its non-locality. Indeed, the form of Equation (5) depends on the global geometry of $\Gamma$, although, in practice, this dependence reduces to a neighborhood of $\widetilde{c}$ the size of which is determined by $d(\widetilde{c}, \Gamma)$ and $\rho$. This non-local behavior is more apparent when the solution of Equation (5) is explicitly presented as the limit of an iterative optimization process

---

[3]https://codeparade.itch.io/marblemarcher

[4]For sufficiently large $\rho$, no unique $p_t$ exists. A realistic model for rigid body motion must integrate the downward force from all (the potentially uncountably many) contact points.

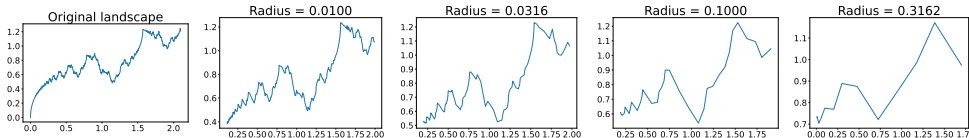

Figure 2: Center trajectory of RBO with different radii on the Riemann function (100$^{\text{th}}$ partial sum).

$p_{t+1} = \lim_{k \to +\infty} p_{t+1}^{(k)}$, where $\left( p_{t+1}^{(k)} \right)_{k \geq 0}$ is given by:

$$
\begin{aligned}
p_{t+1}^{(k)} &= \left( \theta^{(k)}, f\left( \theta^{(k)} \right) \right) \\
\theta^{(k+1)} &= \theta^{(k)} - \gamma \, \nabla d^2(\cdot, \widetilde{c})|_{\theta^{(k)}} \\
&= \theta^{(k)} - \gamma \left[ \theta^{(k)} - \widetilde{\theta} + \left( f\left( \theta^{(k)} \right) - \widetilde{y} \right) \nabla f\left( \theta^{(k)} \right) \right],
\end{aligned}
\tag{6}
$$

where $(\widetilde{\theta}, \widetilde{y}) := \widetilde{c}$. The dynamics of Equation (6), and by extension, its limit, $p_{t+1}$, depend on $f$'s zeroth and first order behavior on a much larger set of points than just $\{p_t\}$, as is the case for spatially local optimizers.

## 3 THEORETICAL ANALYSIS

---

**Algorithm 1:** Rolling ball optimizer

---

**Input:** Loss function $f : \mathbb{R}^d \to \mathbb{R}$; Learning rate $\eta > 0$; Ball radius $\rho > 0$; Initial solution $\theta_0 \in \mathbb{R}^d$; Number of iterations $T \geq 1$

1 **begin**
2  $\quad p_0 \leftarrow (\theta_0, f(\theta_0))$
3  $\quad c_0 \leftarrow p_0 + \rho\nu(p_0)$
4  $\quad$ **for** $t = 0$ **to** $T$ **do**
5  $\quad\quad \widetilde{c}_{t+1} \leftarrow c_t - \eta\tau(p_t)$
6  $\quad\quad p_{t+1} \leftarrow \text{proj}_\Gamma \widetilde{c}_{t+1}$
7  $\quad\quad c_{t+1} \leftarrow p_{t+1} + \rho\nu(p_{t+1})$
8  $\quad$ **end**
9  $\quad (\theta^*, f(\theta^*)) \leftarrow p_T$
10 **end**

**Output:** Optimal solution $\theta^*$

---

The central claim of this paper is that RBO is less sensitive to the fine-grained topography of the loss landscape than its point-like counterparts. Indeed, it is intuitive that the trajectory of a car's wheels' axles ($\rho \sim 1\text{m}$) does not respond to a change in the geometry of the road on the Planck scale, nor does a container ship ($\rho \sim 100\text{m}$) feel waves of length $\lambda \sim 1\text{m}$, which a surfboard would. This intuition can be summarized as: bodies do not feel landscape changes much smaller than their own size, and larger bodies can ignore larger changes than smaller ones. In the context of ball rolling motion, this intuition dictates that, balls of a "large" radius ($\rho$) cannot distinguish between two landscapes, so long as they differ by a sufficiently "small" (compared to $\rho$) amount. This is exceedingly useful, as it allows us to pull the following trick. If RBO is well-behaved on a simple landscape $\Gamma$, and if $\rho$ is large enough that $\Gamma$ is indistinguishable from some other, possibly more complex landscape $\Gamma'$, we can infer many properties of the dynamics of RBO on $\Gamma'$ from their well-understood counterparts on $\Gamma$. Figure 2 illustrates this effect in the case of the *Riemann function* $f : \mathbb{R} \to \mathbb{R}$, defined by: $f(\theta) = \sum_{n=1}^{\infty} \frac{1}{n^2} \sin\left( n^2\theta \right)$.

It is apparent that as $\rho$ increases, the smallest fluctuations in the loss function are smoothed out first, followed by larger fluctuations, until only the largest scale shape of the curve is preserved. We refer to this effect as the *ironing property* of RBO. In this section, we rigorously formulate the ironing property, which we prove in Appendix A. The fundamental assumption we will be using is that the invariant in Equation (1) holds. Under the same assumption, we show that, unlike point-like optimizers, some points of $\Gamma$ can never be reached by RBO, resulting in another form of invariance: invariance to unreachable regions.

### 3.1 Notation and conventions

Throughout this section we will denote by $B(p, \rho)$ for $\rho > 0$ and $p \in E$, where $E$ is some normed vector space (in our case, either $\mathbb{R}^d$ or $\mathbb{R}^{d+1}$, which will be clear from context), the *open* ball of radius $r$ centered at $p$. That is

$$B(p, \rho) := \{x \in E | \|x - p\| < \rho\}.$$

Similarly, we denote by $S(p, \rho)$ the sphere of radius $\rho$ centered at $p$. That is $S(p, \rho) = \partial B(p, \rho)$, the boundary of $B(p, \rho)$. We overload our notation for subsets $X \subset E$ in the following way

- For $p \in E$, $d(p, X) := \inf \{\|x - p\| \mid x \in X\}$, as is customary.
- For $B(X, \rho) = \{x \in E | d(x, X) < \rho\} = \bigcup_{x \in X} B(x, \rho)$.
- In the particular case where $E = \mathbb{R}^{d+1}$, and $X$ is the graph of a function $\mathbb{R}^d \to \mathbb{R}$, we also define $S(X, \rho)$ as the upper connected component of $\partial B(X, \rho)$.

$B(X, \rho)$ and $S(X, \rho)$ are called the $\rho-$tubular neighborhood of $X$ and the $\rho-$offset (manifold) of $X$ respectively. Finally, for $X, Y \subset E$, we define the *Hausdorff distance* as

$$d_H(X, Y) := \inf \{\varepsilon > 0 | X \subset B(Y, \varepsilon) \wedge Y \subset B(X, \varepsilon)\}.$$

### 3.2 The ironing property

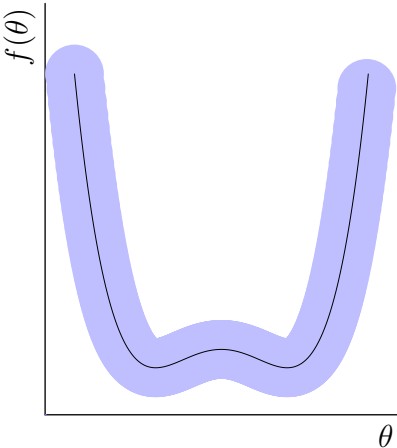

A natural way to formalize the intuition outlined in the first paragraph of Section 3 is to consider a bounded function $\varphi : \mathbb{R}^d \to \mathbb{R}$ with graph $\Gamma$, and examine the locus of centers of all spheres with radius $\rho$ that are tangent to $\Gamma$ (from above). This turns out to be exactly the offset manifold $S(\Gamma, \rho)$. Noting that $S(\Gamma, \rho)$ itself is the graph of a function, defined by[5]

$$\varphi_\rho(\theta) = \sup \left\{ y \in \mathbb{R} \ \middle| \ \exists \theta' \in \mathbb{R}^d, \left\|\theta' - \theta\right\|^2 + \left(y - \varphi\left(\theta'\right)\right)^2 < \rho^2 \right\}, \tag{7}$$

An exact formulation of this intuition is the assertion that $\varphi_\rho$ approaches a constant as $\rho \to +\infty$. This is essentially the statement of Lemma 1.

**Leamma 1** (Weak ironing).
*For every continuous, bounded function $\varphi : \mathbb{R}^d \to \mathbb{R}$, $\varphi_\rho - \rho \to \sup \varphi$ locally uniformly as $\rho \to +\infty$.*

While Lemma 1 is a faithful translation of the previously provided intuition on the physical motion of cars on roads, or ships on waterways, both of which are globally flat —although locally oscillatory—

---

[5]It is straightforward to show that, as the radius $\rho \to 0$, $\varphi_\rho$ converges to $\varphi$ locally uniformly. That is, RBO converges to SGD in the limit of infinitesimal radius.

surfaces, the assumption of boundedness is too strong for use in practical settings, as, with little exception, loss functions are unbounded. That being said, weak ironing is not as useless as one might believe at first glance. For instance, it has the interesting consequence that a sufficiently large ball rolling over $\Gamma$ behaves as if it were rolling over a horizontal hyperplane, moving in the horizontal direction (assuming a horizontal component of velocity). Contrast this with a point-particle like gradient descent, which would get stuck in the first minimum it encountered. More importantly, it serves as a stepping stone for stronger, more immediately useful results, like Proposition 2.

**Proposition 2** (Linear ironing). *Let $f : \mathbb{R}^d \to \mathbb{R}, \theta \mapsto \langle a, \theta \rangle + b$ for some $a, b \in \mathbb{R}^d$ be an affine function, and let $\varphi : \mathbb{R}^d \to \mathbb{R}$ be a bounded continuous function. Furthermore, let $\Gamma$ and $\Gamma'$ be the graphs of $f$ and $f + \varphi$, respectively. Then, for any compact $K \subset \Gamma$ (compact in the topology of $\Gamma$, identified with $\mathbb{R}^d$), one has*

$$\mathrm{d_H}\Big(\mathrm{S}(\Gamma, \rho) \cap K^\perp, \mathrm{S}(\Gamma', \rho) \cap K^\perp\Big) \to 0$$

*as $\rho \to +\infty$, where $K^\perp = K + \nu\mathbb{R}$ and $\nu = (a, -1)$. In particular, for any $\varepsilon > 0$, there exists $\rho_0 > 0$ such that, for any $\rho > \rho_0$,*

$$\mathrm{S}(\Gamma, \rho) \cap K^\perp \subset \mathrm{B}\Big(\mathrm{S}(\Gamma, \rho) \cap K^\perp, \varepsilon\Big)$$

Unlike weak ironing, linear ironing applies to a non-trivial if unrealistic setting. Since affine functions have (constant) non-zero gradient, they can be subjected to gradient descent (and to RBO). While the dynamics of SGD are at the mercy of $\varphi$ and its derivatives, Proposition 2 ensures that RBO descends $\Gamma'$ just as surely as it does $\Gamma$, provided that $\rho$ is large enough. A final note to make is that ironing does not apply only to bounded vertical shifts. In fact, the only requirement for the conclusion to follow is that $\Gamma' \subset \mathrm{B}(\Gamma, M)$ for some $M < +\infty$. In particular, this applies to bounded horizontal shifts (and indeed, bounded shifts in any direction), which for non-affine functions are not equivalent. This implies that as $\rho$ grows, RBO doesn't forget only the fine structure of $\Gamma$, but the exact location of its starting point, since initializing at $p$ is equivalent to initializing at $q$ and shifting $\Gamma$ by $p - q$.

In the fully general case of an arbitrary continuous function $f : \mathbb{R}^d \to \mathbb{R}$, it stands to reason that the ironing property still holds. However, a proof of Conjecture. 3 still eludes us. Further refinement of the ironing property, and an eventual proof in the general case, are deferred to future work.

**Conjecture 3** (Strong ironing property). *Let $f, g : \mathbb{R}^d \to \mathbb{R}$ be two continuous functions, and let $\Gamma, \Gamma'$ be their respective graphs. If $\mathrm{d_H}(\Gamma', \Gamma) < +\infty$, then, $\mathrm{S}(\Gamma, \rho)$ and $\mathrm{S}(\Gamma', \rho)$ are $\varepsilon_\rho-$almost isometric, with $\varepsilon_\rho \to 0$ as $\rho \to +\infty$.*

### 3.3 UNREACHABLE POINTS

In Section 3.2, we established that RBO has a smoothing effect through purely static geometric reasoning. In this section, we examine one of the dynamical constraints that can be equivalently seen as leading to, or arising from the ironing property: the existence of unreachable points. Similarly to the rolling ball algorithm from image processing (Sternberg, 1983), RBO rolls over the smooth regions of the loss landscape, but is unable to penetrate sharp valleys (compared to its radius). As such, it is naturally invariant to changes in the loss landscape that only affect unreachable points. In fact, since the set of unreachable points is open, it is also invariant to changes that only affect a small region around an unreachable point. In this section, we define unreachable points, provide formal results showing that sharp local minima are unreachable for sufficiently large $\rho$, and that an unreachable point has an open neighborhood of unreachable points. Proofs of these results can be found in Appendix A.

We start by formalizing the definition of an unreachable point. Intuitively, a point is unreachable by a ball of radius $\rho$, if the ball has to go through the loss landscape in order to reach it. Definition 1 puts this intuition in symbols.

**Definition 1** (Unreachable point).
Let $f : \mathbb{R}^d \to \mathbb{R}$ be a continuous function and let $\Gamma$ be its graph. A point $p \in \Gamma$ is said to be $\rho$-unreachable ($\rho > 0$) if and only if for all possible centers $c \in \mathrm{epi}\, f$,

$$\|c - p\| = \rho \implies \exists q \in \Gamma, \|c - q\| < \rho.$$

The notion of unreachability can be used to define a notion of sharpness for local minima. Proposition 4 states that all local minima are unreachable for sufficiently large $\rho$. The sharp ones are the ones unreachable for smaller $\rho$. By linking this to the Hessian spectral norm, Proposition 4 implies that this notion of sharpness coincides with well-known formulations of the concept (Keskar et al., 2022; Dinh et al., 2017)

**Proposition 4** (Unreachable sharp minima).
*Let $f : \mathbb{R}^d \to \mathbb{R}$ be a $\mathcal{C}^2$ function and let $\Gamma$ be its graph. Furthermore, let $p = (\theta_0, f(\theta_0)) \in \Gamma$ be a local minimum of $f$. Finally, define the* sharpness *at $p$ as $\sigma = \left\| \partial^2 f(\theta_0) \right\|$. Then, for all $\rho > \frac{1}{\sigma}$, $p$ is $\rho$-unreachable.*

**Leamma 5** (Non-isolated unreachable points).
*Let $f : \mathbb{R}^d \to \mathbb{R}$ be a continuous function, $\Gamma$ be its graph, and $\rho > 0$. If a point $p \in \Gamma$ is $\rho$-unreachable, then, there exists an $\varepsilon > 0$ such that every point in $\mathrm{B}(p, \varepsilon) \cap \Gamma$ is also $\rho$-unreachable. In other words, the set of $\rho$-unreachable points is open in $\Gamma$.*

## 4 EXPERIMENTAL RESULTS

To evaluate RBO and gain concrete insights into its behavior, we first compare it with baseline optimizers—SGD, Entropy-SGD, and SAM—in terms of performance and training dynamics across a range of image classification tasks. We then analyze the impact of the radius on its optimization behavior.

### 4.1 EXPERIMENTAL SETUP

All experiments are conducted on a single NVIDIA RTX 4050 GPU with 6 GB of VRAM. For RBO, we used a radius $\rho = 1$ and a learning rate $\eta = 6$. For the projection solver, we used SGD with a learning rate $\gamma = 0.1$ for 20 steps. A learning rate of $\eta = 0.01$ was used for SGD. As for SAM and Entropy-SGD, we used the recommended hyperparameters of $\rho = 0.05$ for the former, and $\gamma = 3 \times 10^{-3}$ with 20 Langevin steps for the latter. For each optimizer, the same hyperparameters are used for all model/dataset combinations. Our implementation of Entropy-SGD is adapted from an open-source implementation [6].

### 4.2 PERFORMANCE COMPARISON

We use three standard image classification datasets in our experiments: MNIST, CIFAR-10, and CIFAR-100. On the former, we train a MultiLayer Perceptron (MLP) withe two hidden layers of size 256 each, ResNet-6 He et al. (2016) and VGG-9 Simonyan & Zisserman (2015). As for the other two datasets, we train ResNet-8 and VGG-9. All runs used 10 epochs of training. We report the best loss and accuracy on the validation set, averaged over 5 independent runs, along with $95\%$ confidence intervals for each metric in Table 1.

| Model | SGD | | Entropy-SGD | | SAM | | RBO (our method) | |
|---|---|---|---|---|---|---|---|---|
| | Loss | Accuracy | Loss | Accuracy | Loss | Accuracy | Loss | Accuracy |
| MNIST | | | | | | | | |
| MLP | $0.291 \pm 0.002$ | $91.77\% \pm 0.09\%$ | $0.157 \pm 0.002$ | $95.22\% \pm 0.19\%$ | $0.093 \pm 0.001$ | $97.22\% \pm 0.14\%$ | $\mathbf{0.080 \pm 0.002}$ | $\mathbf{97.51\% \pm 0.14\%}$ |
| ResNet-6 | $0.094 \pm 0.002$ | $97.59\% \pm 0.08\%$ | $0.056 \pm 0.004$ | $98.18\% \pm 0.15\%$ | $0.028 \pm 0.002$ | $\mathbf{99.11\% \pm 0.12\%}$ | $\mathbf{0.027 \pm 0.001}$ | $99.07\% \pm 0.08\%$ |
| VGG-9 | $0.046 \pm 0.001$ | $98.78\% \pm 0.04\%$ | $0.046 \pm 0.001$ | $98.57\% \pm 0.06\%$ | $\mathbf{0.018 \pm 0.001}$ | $\mathbf{99.39\% \pm 0.06\%}$ | $0.021 \pm 0.001$ | $99.27\% \pm 0.05\%$ |
| CIFAR-10 | | | | | | | | |
| ResNet-8 | $1.206 \pm 0.009$ | $56.54\% \pm 0.59\%$ | $1.363 \pm 0.028$ | $59.16\% \pm 0.74\%$ | $\mathbf{0.885 \pm 0.014}$ | $69.09\% \pm 0.28\%$ | $1.538 \pm 0.065$ | $\mathbf{71.58\% \pm 0.90\%}$ |
| VGG-9 | $0.964 \pm 0.014$ | $66.04\% \pm 0.48\%$ | $1.082 \pm 0.017$ | $65.46\% \pm 0.64\%$ | $\mathbf{0.666 \pm 0.026}$ | $77.81\% \pm 0.58\%$ | $0.856 \pm 0.046$ | $\mathbf{81.87\% \pm 0.60\%}$ |
| CIFAR-100 | | | | | | | | |
| ResNet-8 | $3.452 \pm 0.013$ | $19.28\% \pm 0.30\%$ | $3.025 \pm 0.016$ | $28.33\% \pm 0.47\%$ | $\mathbf{2.479 \pm 0.029}$ | $36.26\% \pm 0.58\%$ | $3.787 \pm 0.059$ | $\mathbf{37.11\% \pm 0.98\%}$ |
| VGG-9 | $2.972 \pm 0.040$ | $29.37\% \pm 0.76\%$ | $3.001 \pm 0.022$ | $28.98\% \pm 0.95\%$ | $\mathbf{1.986 \pm 0.023}$ | $47.17\% \pm 0.54\%$ | $2.836 \pm 0.032$ | $\mathbf{50.07\% \pm 0.79\%}$ |

Table 1: Comparing the test performance of RBO against other optimizers.

The results show with clarity that RBO and SAM have better generalization performance than SGD and Entropy-SGD. Between the two, RBO outperforms globally in terms of accuracy, while SAM maintains the advantage in terms of cross-entropy loss. We note that the hyperparameters of RBO

---

[6]https://github.com/steph1793/Entropy-SGD.git

have not been tuned for any of the experiments, and as a result, the results in Table 1 are not representative of its best case performance.

### 4.3 TRAINING DYNAMICS COMPARISON

Having compared RBO's generalization performance to that of the other optimizers we have considered, we now examine its optimization behavior. Figures 3, 4a and 4b show the learning curves of all four optimizers on the MNIST, CIFAR-10, and CIFAR-100 datasets, respectively. In every case, RBO achieves lower training loss and higher training accuracy than any other optimizer we have evaluated. Furthermore, it is consistently the fastest to converge, and has the lowest limiting cross-entropy loss. This is particularly apparent on the CIFAR-10/100 datasets (Figure 4), where it

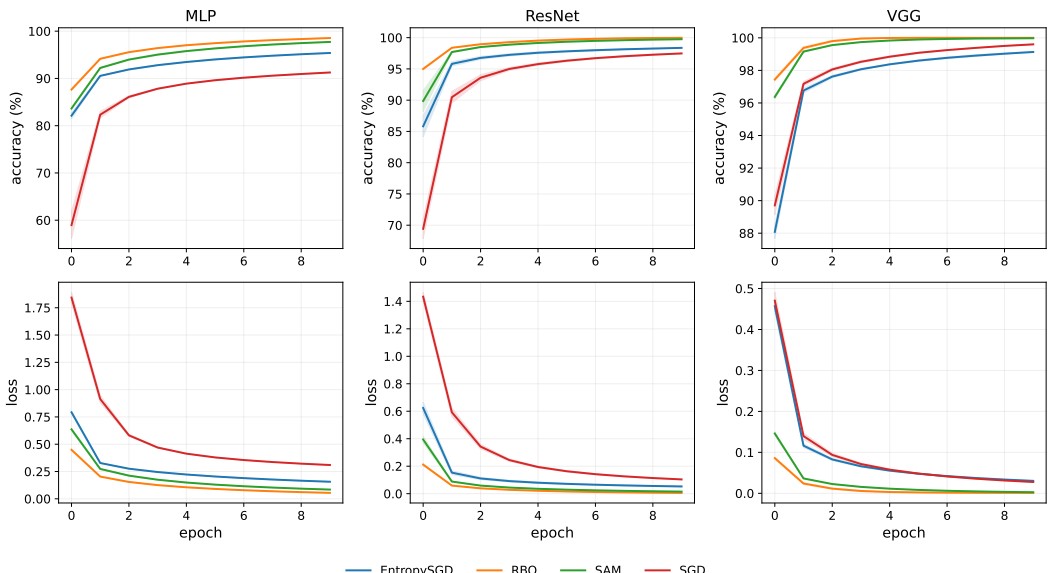

Figure 3: Training curves for the MNIST dataset.

beats the final training performance of the other optimizers in half the number of epochs. Compared

(a) CIFAR-10                    (b) CIFAR-100

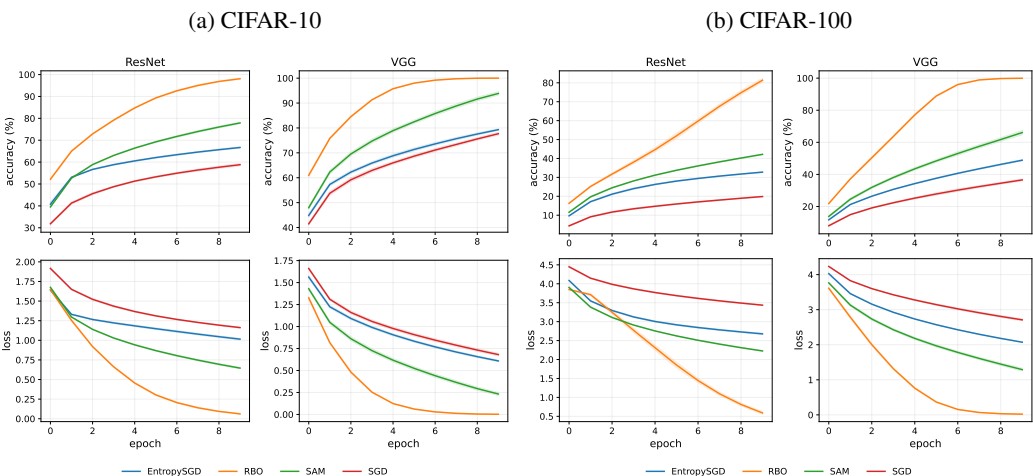

Figure 4: Training curves for the CIFAR-10 and CIFAR-100 datasets.

to SGD in particular, RBO overtakes its final training performance in one or two epochs on every

dataset/model combination. This is, in large part, due to the fact that RBO can converge with extremely large learning rates (Section 4.4), and can therefore claim the speed benefits of large step sizes without the stability cost. The validation performance of RBO does not seem to display a similarly strong advantage. The reasons for this should be the subject of future investigation.

### 4.4 Effect of the radius

The behavior of RBO is governed by the scale of its interaction with the loss landscape, which, in turn, is determined by the radius $\rho$. All the unique properties of RBO depend on this hyperparameter, which makes understanding its effect on the dynamics critical for understanding and correctly using RBO.

Figure 5 shows the validation accuracy of an MLP trained with RBO for three epochs on the MNIST dataset against a range of values of $\rho$ and $\eta$. The radii are log-spaced between 0.1 and 10, and for each radius $\rho$, the learning rates are log-spaced between $0.01\rho$ and $10\rho$. We opted for this coupling instead of a grid search to reduce the search space, since using large learning rates with small radii leads to divergent dynamics. We observe that performance seems to increase monotonically with both $\rho$ and $\eta$, and that training is stable with $\eta$ as large as 100. This trend cannot continue indefinitely, as, for example, with a radius of 30, and a learning rate of 100, the values of the loss function diverged very quickly. A reasonable conjecture seems to be that RBO undergoes two phase transitions: first, from the microscopic

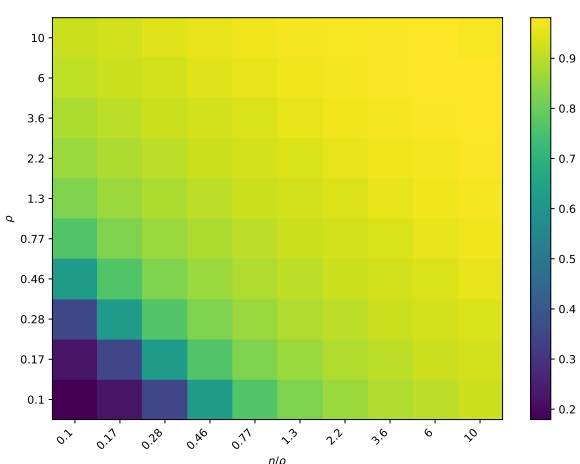

Figure 5: Final validation accuracy of an MLP trained with RBO on the MNIST dataset as a function of $\rho$ and $\eta$.

regime where $\rho$ is small enough for point-like dynamics to be a good approximation, and then again, from the macroscopic regime to a *"megascopic"* regime, where $\rho$ is too large for the dynamics to stabilize. Determining the critical scales of these transitions (and confirming that they exist at all) is a future research direction of particular importance to tuning RBO.

## 5 Conclusion, limitations, and future work

The point-like nature of the dominant family of optimizers leads to several undesirable properties when the loss landscape is complex, as is often the case in DL. To provide performance guarantees in the presence of challenging geometries, a non-local update rule is required. In this paper, we propose one (arguably, the simplest) such update rule, simulating the motion of a rigid sphere on the loss landscape. The resulting optimizer has a smoothing effect on the loss landscape, is approximately invariant to small perturbations, and is stable with larger learning rates.

While RBO and non-local optimizers in general are a promising candidate for a truly robust training procedure for DNNs, much about them remains to be understood, and they are not without their shortcomings. First, and most pressingly, non-local optimizers, including RBO and Entropy-SGD, tend to be significantly more computationally expensive than standard gradient-based optimizers. This complicates large-scale adoption and experimentation. Second, using an optimization problem as an update rule introduces errors arising from the approximate nature of the solver. These errors can accumulate over time, and are of particular concern given the chaotic early dynamics of RBO with large radii. It is unclear how the dimensionality of the problem influences these errors, but it is not unlikely that the curse of dimensionality exacerbates this problem.

Finally, this work leaves a number of important questions open. Chief among them is rigorously establishing the ironing property in the general case. Other equally important open questions include the effect RBO has on other dimensions of learning, like fairness and robustness to Byzantine attacks in federated settings. Further experimental evaluation is also required to adequately understand the behavior of RBO in realistic scenarios.

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

# A   PROOFS OF THEORETICAL RESULTS

**Leamma 1** (Weak ironing). *For every continuous, bounded function $\varphi : \mathbb{R}^d \to \mathbb{R}$, $\varphi_\rho - \rho \to \sup \varphi$ locally uniformly as $\rho \to +\infty$.*

*Proof.* Throughout this proof, we will make use of the equivalent expression for $\varphi_\rho$:

$$
\begin{aligned}
\varphi_\rho(\theta) &= \sup_{\|\theta' - \theta\| < \rho} \left[ \varphi\left(\theta'\right) + \sqrt{\rho^2 - \|\theta' - \theta\|^2} \right] \\
&= \sup_{\|s\| < \rho} \left[ \varphi\left(\theta + s\right) + \sqrt{\rho^2 - \|s\|^2} \right].
\end{aligned}
\tag{8}
$$

We start by showing pointwise convergence to $M := \sup \varphi$. Fixing $\theta \in \mathbb{R}^d$, and $\varepsilon > 0$, we find a $\theta' \in \mathbb{R}^d$ such that $M - \varphi(\theta') < \varepsilon/2$. For any $\rho > \|\theta' - \theta\|$, we have

$$
\varphi_\rho(\theta) \geq f(\theta') + \sqrt{\rho^2 - \|\theta' - \theta\|^2} \geq M - \frac{\varepsilon}{2} + \sqrt{\rho^2 - \|\theta' - \theta\|^2},
$$

which implies that

$$
\begin{aligned}
\varphi_\rho(\theta) - \rho &\geq M - \frac{\varepsilon}{2} + \sqrt{\rho^2 - \|\theta' - \theta\|^2} - \rho \\
&= M - \frac{\varepsilon}{2} + \rho \left( \sqrt{1 - \frac{\|\theta' - \theta\|^2}{\rho^2}} - 1 \right) \\
&= M - \frac{\varepsilon}{2} - \frac{1}{\rho} \frac{\|\theta' - \theta\|^2}{1 + \sqrt{1 - \frac{\|\theta' - \theta\|^2}{\rho^2}}} \\
&\geq M - \frac{\varepsilon}{2} - \frac{\|\theta' - \theta\|^2}{\rho}.
\end{aligned}
$$

For $\rho > 2\|\theta' - \theta\|^2/\varepsilon$, this yields $\varphi_\rho(\theta) - \rho > M - \varepsilon$, which, combined with the trivial inequality $\varphi_\rho(\theta) \leq M + \rho$, implies the desired result.

To establish locally uniform convergence, let $K \subset \mathbb{R}^d$ be compact, and let $A = \sup_{\theta \in K} \|\theta' - \theta\|$, which is finite by the extreme value theorem. For any $\varepsilon > 0$ and $\rho > 2A^2/\varepsilon$, the previous argument shows that

$$
\forall \theta \in K, |\varphi_\rho(\theta) - \rho - M| < \varepsilon,
$$

or, equivalently, $\left\| \varphi_{\rho|_K} - \rho \right\|_\infty < \varepsilon$, the desired conclusion. $\square$

**Proposition 2** (Linear ironing). *Let $f : \mathbb{R}^d \to \mathbb{R}, \theta \mapsto \langle a, \theta \rangle + b$ for some $a, b \in \mathbb{R}^d$ be an affine function, and let $\varphi : \mathbb{R}^d \to \mathbb{R}$ be a bounded continuous function. Furthermore, let $\Gamma$ and $\Gamma'$ be the graphs of $f$ and $f + \varphi$, respectively. Then, for any compact $K \subset \Gamma$ (compact in the topology of $\Gamma$, identified with $\mathbb{R}^d$), one has*

$$
d_H \left( S(\Gamma, \rho) \cap K^\perp, S(\Gamma', \rho) \cap K^\perp \right) \to 0
$$

*as $\rho \to +\infty$, where $K^\perp = K + \nu \mathbb{R}$ and $\nu = (a, -1)$. In particular, for any $\varepsilon > 0$, there exists $\rho_0 > 0$ such that, for any $\rho > \rho_0$,*

$$
S(\Gamma, \rho) \cap K^\perp \subset B \left( S(\Gamma, \rho) \cap K^\perp, \varepsilon \right)
$$

*Proof.* First, we need the following lemma:

**Corollary 3.** *Let $\Phi : \mathbb{R}^d \to \mathbb{R}^{d+1}$ be a continuous function, such that $\left\| \theta \mapsto \langle \Phi(\theta), e_{d+1} \rangle \right\|_\infty < +\infty$, that is, the last component of $\Phi$ is bounded. In particular, let $M = \sup \langle \Phi(\theta), e_{d+1} \rangle$. Furthermore, let $\Gamma = \Phi(\mathbb{R}^d)$, and assume that $\Gamma$ intersects every vertical line. Then, for every compact $K \subset \mathbb{R}^d$,*

$$\mathrm{d_H}\big( \mathrm{S}(\Gamma, \rho) \cap (K \times \mathbb{R}), K \times \{\rho + M\} \big) \to 0$$

*as $\rho \to +\infty$.*

*Proof.* Defining

$$\varphi(\theta) := \sup \left\{ y \in \mathbb{R} \ \middle| \ \exists \vartheta \in \mathbb{R}^d, \Phi(\vartheta) = (\theta, y) \right\},$$

one verifies that $\mathrm{S}(\Gamma, \rho)$ is the graph of

$$\varphi_\rho(\theta) = \sup_{\|s\| < \rho} \left[ \varphi(\theta + s) + \sqrt{\rho^2 - \|s\|^2} \right],$$

and that $\varphi$ is bounded. The result then follows immediately from Lemma 1[7]    $\square$

Now, proceed with proving Proposition 2. Let $T : \mathbb{R}^{d+1} \to \mathbb{R}^{d+1}$ be a rigid transformation such that $T(\Gamma) = \mathbb{R}^d \times \{0\}$, and $\langle T_* \nu, e_{d+1} \rangle > 0$. Defining $\Phi : \mathbb{R}^d \to \mathbb{R}^{d+1}$ such that $\Phi(\mathbb{R}^d) = T(\Gamma')$, we see immediately that the conditions of Corollary 3 are met. As such, for any compact $K \subset \mathbb{R}^d$,

$$\mathrm{d_H}\Big( \mathrm{S}(T(\Gamma'), \rho) \cap (K \times \mathbb{R}), \mathrm{S}(T(\Gamma), \rho) \cap (K \times \mathbb{R}) \Big) = \mathrm{d_H}\Big( \mathrm{S}(T(\Gamma'), \rho) \cap (K \times \mathbb{R}), K \times \{\rho + M\} \Big) \to 0$$

as $\rho \to +\infty$. Applying $T^{-1}$, and noting that $T^{-1}(K \times \mathbb{R}) = K^\perp$ we get

$$\mathrm{d_H}\Big( \mathrm{S}(\Gamma, \rho) \cap K^\perp, \mathrm{S}(\Gamma', \rho) \cap K^\perp \Big) \to 0$$

as $\rho \to +\infty$, concluding the proof.    $\square$

Unlike weak ironing, linear ironing applies to a non-trivial if unrealistic setting. Since affine functions have (constant) non-zero gradient, they can be subjected to gradient descent (and to RBO). While the dynamics of SGD are at the mercy of $\varphi$ and its derivatives, Proposition 2 ensures that RBO descends $\Gamma'$ just as surely as it does $\Gamma$, provided that $\rho$ is large enough. A final note to make is that ironing does not apply only to bounded vertical shifts. In fact, the only requirement for the conclusion to follow is that $\Gamma' \subset \mathrm{B}(\Gamma, M)$ for some $M < +\infty$. In particular, this applies to bounded horizontal shifts (and indeed, bounded shifts in any direction), which for non-affine functions are not equivalent. This implies that as $\rho$ grows, RBO doesn't forget only the fine structure of $\Gamma$, but the exact location of its starting point, since initializing at $p$ is equivalent to initializing at $q$ and shifting $\Gamma$ by $p - q$.

In the fully general case of an arbitrary continuous function $f : \mathbb{R}^d \to \mathbb{R}$, it stands to reason that the ironing property still holds. However, a proof of Conjecture. 3 still eludes us. Further refinement of the ironing property, and eventual proof in the general case, are deferred to future work.

**Conjecture 4** (Strong ironing property)**.** *Let $f, g : \mathbb{R}^d \to \mathbb{R}$ be two continuous functions, and let $\Gamma, \Gamma'$ be their respective graphs. If $\mathrm{d_H}\Gamma', \Gamma < +\infty$, then, $\mathrm{S}(\Gamma, \rho)$ and $\mathrm{S}(\Gamma', \rho)$ are $\varepsilon_\rho$−almost isometric, with $\varepsilon_\rho \to 0$ as $\rho \to +\infty$.*

**Proposition 5** (Unreachable sharp minima)**.** *Let $f : \mathbb{R}^d \to \mathbb{R}$ be a $\mathcal{C}^2$ function and let $\Gamma$ be its graph. Furthermore, let $p = (\theta_0, f(\theta_0)) \in \Gamma$ be a local minimum of $f$. Finally, define the* sharpness *at $p$ as $\sigma = \left\| \partial^2 f(\theta_0) \right\|$. Then, for all $\rho > \frac{1}{\sigma}$, $p$ is $\rho$-unreachable.*

*Proof.* Let $S$ be a sphere of radius $\rho > \sigma$ tangent to $\Gamma$ at $p$. The lower hemisphere of $S$ is the graph of a function $g(\theta) = f(\theta_0) + \rho - \sqrt{\rho^2 - \|\theta - \theta_0\|^2}$. One needs only to show that there exists a $\theta'$ such that $\|\theta' - \theta_0\| < \rho$ and $g(\theta') < f(\theta')$, because then, one would have

$$\Big( f(\theta_0) + \rho - f(\theta') \Big)^2 + \left\| \theta' - \theta_0 \right\|^2 < \rho^2,$$

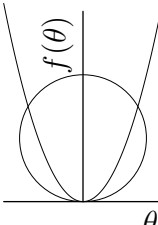

Figure 6: Every local minimum is eventually unreachable.

so $(\theta', f(\theta'))$ is in the open ball bounded $S$, and thus $p$ is unreachable (see Figure 6). To find such a point, let, $u \in \mathbb{R}^d$ be an eigenvector of $\partial^2 f(\theta_0)$ associated with the eigenvalue $\sigma$, such that $\|u\| < \rho$, and expand $f(\theta_0 + u)$ as

$$f(\theta_0 + u) = f(\theta_0) + \frac{1}{2} u^\mathsf{T} \, \partial^2 f(\theta_0) u + \mathcal{O}\left(\|u\|^3\right)$$

$$= f(\theta_0) + \frac{1}{2}\sigma\|u\|^2 + \mathcal{O}\left(\|u\|^3\right),$$

and similarly expand $g(\theta_0 + u)$ as

$$g(\theta_0 + u) = f(\theta_0) + \rho - \sqrt{\rho^2 - \|u\|^2}$$

$$= f(\theta_0) + \frac{1}{2\rho}\|u\|^2 + \mathcal{O}\left(\|u\|^3\right).$$

We get that the difference

$$f(\theta_0 + u) - g(\theta_0 + u) = \frac{1}{2}\left(\sigma - \frac{1}{\rho}\right)\|u\|^2 + \mathcal{O}\left(\|u\|^3\right),$$

which is positive for small $\|u\|$ by the fact that $\sigma > \frac{1}{\rho}$. $\qquad\square$

**Leamma 6** (Non-isolated unreachable points). *Let $f : \mathbb{R}^d \to \mathbb{R}$ be a continuous function, $\Gamma$ be its graph, and $\rho > 0$. If a point $p \in \Gamma$ is $\rho$-unreachable, then, there exists an $\varepsilon > 0$ such that every point in $\mathrm{B}(p, \varepsilon) \cap \Gamma$ is also $\rho$-unreachable. In other words, the set of $\rho$-unreachable points is open in $\Gamma$.*

*Proof.* We assume that at least one point in $\Gamma$ is $\rho$-unreachable, otherwise, the result is trivial. One shows with ease that $p \in \mathbb{R}^{d+1}$ is $\rho$-unreachable if and only if $\mathrm{S}(p, \rho) \subset \mathrm{B}(\Gamma, \rho)$. It is therefore sufficient to prove that if $\mathrm{B}(\Gamma, \rho)$ contains $\mathrm{S}(p, \rho)$, then it must contain all the spheres $\mathrm{S}(q, \rho)$ with $q \in \Gamma$ and $\|q - p\| < \varepsilon$ for some $\varepsilon > 0$.

The key observation for this proof is the fact that $\mathrm{B}(\Gamma, \rho)$ is open. Therefore, its complement, $E = \mathbb{R}^{d+1} \setminus \mathrm{B}(\Gamma, \rho)$ is closed. It follows that, for any unreachable point $p \in \Gamma$,

$$d\left(\mathrm{S}(p, \rho), E\right) := \inf\left\{\|x - y\| \mid x \in E \wedge y \in \mathrm{S}(p, \rho)\right\} > 0,$$

since otherwise, $\mathrm{S}(p, \rho) \cap E \neq \varnothing$, because $\mathrm{S}(p, \rho)$ is compact and $E$ is closed, contradicting the assumption that $\mathrm{S}(p, \rho) \subset \mathrm{B}(\Gamma, \rho)$. For $\varepsilon = d\left(\mathrm{S}(p, \rho), E\right)/2$, let $q \in \Gamma$ be a point such that $\|q - p\| < \varepsilon$, and let $a \in \mathrm{S}(q, \rho)$.

On one hand, $a + p - q \in \mathrm{S}(p, \rho)$, since

$$\left\|(a + p - q) - p\right\| = \|a - q\| = \rho,$$

and the other hand

$$d(a, \mathrm{S}(q, \rho)) \leq \left\|a - (a + p - q)\right\| = \|p - q\| = \varepsilon,$$

so $a \in \mathrm{B}(\Gamma, \rho)$. Since $a$ was chosen arbitrarily on $\mathrm{S}(q, \rho)$ it follows that $\mathrm{S}(p, \rho) \subset \mathrm{B}(\Gamma, \rho)$, which concludes the proof. $\qquad\square$

---

[7]With a small caveat: $\varphi$ need not be continuous, but the proof of Lemma 1 still yields the same result without modification.

