# OpenReview forum: "Rolling Ball Optimizer: Learning by ironing out loss landscape wrinkles"
_ICLR.cc/2026/Conference — Submitted to ICLR 2026_

### Official Review · Reviewer_T7n4 · 2025-10-28

**Soundness:** 3
**Presentation:** 1
**Contribution:** 2
**Rating:** 2
**Confidence:** 4

**Summary:**

This work proposes a new optimizer: the rlling ball optimizer. It consists in simulating a rolling ball over the loss landscape. This work then present several theoretical evidences of the benefits of this approach, and empirically asses its performance.

**Strengths:**

The algorithm is novel and interesting. The theoretical analysis serves as a solid foundation for the proposed method.

**Weaknesses:**

### Presentation
Lack of details for the algorithm: all the crucial steps of the algorithms 5-6-7 (Alg.1) need more explanations (i.e. for now I could not reproduce the algorithm from the description provided).

### Experiments
The experimental section is quite weak: only small-scale vision datasets are used with small models. Moreover, it seems that hyperparameter tuning is not done for baselines and it is not clear how it is done for the proposed method.

### Minor
- l280: "fromal" -> "formal"

To sum up, though the idea is interesting and the theoretical analysis solid, the presentation and experimental section need significant improvements.

**Questions:**

- Coud you make the relationship between the notion of "Unreachable point" and the trajectory of the algorithm more clear? Is it guaranteed that the algorithm will not visit those points?
- Address the weaknesses mentioned above.

---

> ### Author Response · Authors · 2025-12-01
>
> - **Algorithm unclear.** We have added additional details (particularly, by adding explicit formulas for $\\tau$ and $\\nu$) to the paper, and included a link to a full implementation for reproducibility.
> - **Relatively small benchmarks.** We have limited ourselves to relatively small models and datasets due to hardware and time constraints. All of our experiments have been conducted on a single GPU laptop. While we are working to enrich the experiment suite, this effort is beyond the scope of the discussion period's timeframe.
> - **Hyperparameter tuning.** We have not tuned the baseline optimizers, opting for reasonable defaults that we reused for all experiments. As for our method, we set our learning rate and radius based on the results reported in section 4.4. To maintain fairness, we also used the same values for all experiments.
> - **Unreachable points.** Equation (1) and Definition 1 taken together do imply that RBO never visits unreachable points. Proposition 4 proves that unreachable points exist, and Lemma 5 strengthens this result to guarantee that not only does RBO never reach unreachable minima, it also can't get “too close” to them.

---

### Official Review · Reviewer_AuYt · 2025-10-31

**Soundness:** 3
**Presentation:** 2
**Contribution:** 2
**Rating:** 2
**Confidence:** 3

**Summary:**

This paper proposes a two-step (descent step and constraint projection) optimization method called RBO which is designed to consider the local geometry. The RBO simulates the motion of a rigid $\rho$-sphere over the loss landscape. The authors argue that it improves generalization performance and optimization convergence.

**Strengths:**

- This paper proposes a novel optimization method with intuitive idea of "rolling ball" which can be beneficial not only for optimization and also for generalization.

**Weaknesses:**

- The crucial weakness is the experiment parts.
    - ResNet-6 and VGG-9 are too small. It would be much better if it is scalable to larger neural networks (e.g., WRN-28-10).
    - The accuracy reported in Table 1 is very far from the state-of-the-arts. It doesn't need to be the state-of-the-arts, but at least, CIFAR-10 performance should be over/around 90% (SAM achieved >97% performance according to the SAM paper). It's unclear whether the hyperparameters of SAM are well-tuned for the small neural networks.
    - What is the computational cost (seconds) and time complexity (with respect to the number of parameters) for the RBO compared to SGD? Where is the bottle neck for the RBO computation?

**Questions:**

see weaknesses

---

> ### Author Response · Authors · 2025-12-01
>
> - **Insufficient dataset/model complexity.** We have limited ourselves to relatively small models and datasets due to hardware and time constraints. All of our experiments have been conducted on a single GPU laptop. While we are working to enrich the experiment suite, this effort is beyond the scope of the discussion period's timeframe.
> - **Baselines do not achieve optimal performance.** We have noted the same issue, which is likely due to the fact that we did not tune the hyperparameters of any optimizer. We are currently working on rerunning all of our tests with optimal hyperparameter sets, which, we believe, should rectify this anomaly.
> - **Time complexity.** Counting the total number of forward and backward passes through the network, one epoch of RBO runs in $\\mathcal{O}(B\\times P)$, where $B$ is the number of batches, and $P$ is the number of projection iterations. Standard optimizers run in $\\mathcal{O}(B)$ time. The number of parameters has the same impact on both types of optimizers, since it is a constant multiple of the number of passes.

---

### Official Review · Reviewer_PTNo · 2025-11-01

**Soundness:** 2
**Presentation:** 3
**Contribution:** 2
**Rating:** 4
**Confidence:** 3

**Summary:**

The paper introduces the Rolling Ball Optimizer (RBO), a "non‑local" optimization procedure that simulates a rigid sphere of radius $\rho > 0$ rolling on the graph of the loss $f(\theta)$. Each iteration takes a descent-like step for the ball’s center and then projects back to an offset of the loss surface by finding the closest point on the graph to the provisional center; the new center is set to the contact point plus the outward normal scaled by $\rho$ (Algorithm 1, Fig. 1, pp. 3–4). The intuition is that a finite‑radius body is insensitive to small‑scale “wrinkles,” so RBO “irons out” fine geometry while respecting large‑scale structure. This is formalized via the offset manifold $S(\Gamma,\rho)$ and a sequence of results: a weak smoothing (“ironing”) lemma for bounded functions, a “linear ironing” proposition for affine trends perturbed by bounded noise, and an unreachability result showing sharp local minima become inaccessible once \rho exceeds a curvature‑dependent threshold (pp. 4–6, 12–13). Empirically, on MNIST, CIFAR‑10, CIFAR‑100 with small networks, RBO is compared to SGD, Entropy‑SGD, and SAM. Table 1 (p. 7) reports that RBO often attains higher validation accuracy than the baselines (not always lowest loss), and Figures 3–4 (pp. 7–8) show faster training‑loss reduction. A heat map (Figure 5, p. 8) suggests wide stability regions at large learning rates when $\rho$ is larger.

**Strengths:**

- Originality (concept): Replaces point‑particle dynamics with finite‑radius body dynamics; non‑locality emerges from a projection onto the graph’s offset. This is a clean, physically motivated design space distinct from SAM/Entropy‑SGD. Fig. 2 (p. 4) compellingly visualizes multi‑scale smoothing as $\rho$ increases.
- Quality (math framing): The offset‑manifold viewpoint and the weak/linear ironing results formalize the smoothing intuition; the unreachability proposition links sharpness to curvature via $\|\nabla^2 f\|$ (pp. 5–6, 12–13).
- Clarity (algorithmic idea): Algorithm 1 (p. 4) clearly separates the descent move and the projection; Fig. 1 helps build geometric intuition.
- Significance (potential): If made practical, a radius‑controlled, non‑local optimizer that is stable at larger step sizes could be impactful. Early results (Table 1, Figs. 3–5) indicate faster training and competitive or better validation accuracy on small benchmarks.

**Weaknesses:**

1. **Metric & scaling are not specified or analyzed.** The projection minimizes Euclidean distance in $\mathbb{R}^{d+1}$ between $\tilde c_{t+1}$ and points on the graph $\{(\theta,f(\theta))\}$ (Eq. (3)), which implicitly equates horizontal parameter units and the vertical loss scale. Without a scaling parameter$\lambda$ to balance $\|\theta-\theta_e\|^2 + \lambda^2 (f(\theta)-y_e)^2$, behavior can change drastically under simple transformations (e.g., multiplying the loss by a constant) or parameter re‑scalings. The paper does not discuss scale invariances, sensitivity to reparameterization, or how \rho interacts with loss magnitude. Eq. (4) also reveals this coupling explicitly.
2. **Algorithmic details missing.** The inner projection solver (Eq. (4)) lacks stopping criteria, step size $\gamma$, iteration budget, warm‑start strategy, and failure handling when multiple contact points exist (footnote 4 acknowledges non‑uniqueness for large $\rho$). These choices materially affect stability and cost.
3. **No convergence or complexity guarantees.** Beyond ironing and unreachability, there is no analysis of convergence of the outer loop (even to stationary points of a smoothed objective), nor cost bounds for the inner projection.
4. **Empirical evaluation is too thin to support claims.**
     - Only 10 epochs on CIFAR‑10/100; these regimes underfit typical recipes and can invert method rankings.
     - No wall‑clock or FLOP accounting; RBO performs an inner optimization each step, so “faster to converge in epochs” may still be slower in time.
     - Baselines and tuning: SGD uses a single LR (0.01) with no momentum/decay/schedule; SAM/Entropy‑SGD hyperparameters are “recommended defaults” but learning rates and schedules are unclear, and a single set is used across tasks. This is not competitive practice and likely underestimates strong baselines.
5. **Missing ablations:** No study of inner‑loop iterations/tolerance, projection errors, or radius–learning‑rate couplings beyond a single heat map on MNIST (Fig. 5).
6. **Theory is partial and special‑case.** The main smoothing effect is proven for bounded functions (Lemma 1) and affine trends (Proposition 2). The central “strong ironing” conjecture is left open (p. 5, p. 13), so the key claim remains unproven in the settings that matter operationally.

**Questions:**

1. Metric/scaling: What metric do you intend for Eq. (3)? As written, it is Euclidean in $\mathbb{R}^{d+1}$. Have you explored a scale parameter \lambda so the projection minimizes $\|\theta-\theta_e\|^2 + \lambda^2 (f(\theta)-y_e)^2$? How sensitive is RBO to multiplying the loss by a constant (e.g., switching from CE to CE×10)?
2. Exact formulas for $\tau(p)$ and $\nu(p)$ would be nice: Please specify the closed forms for a graph $\Gamma=\{(\theta,f(\theta))\}$ and show the $\rho\to 0$ limit rigorously reduces to standard GD. (The abstract states this, but I did not find a proof.)
3. Projection step details: How many inner iterations are used, with what \gamma, tolerance, and warm‑start? How often does the projection fail or find different contact points when multiple exist? Please include an ablation of inner iterations vs. accuracy, and wall‑clock costs per step/epoch.
4. How does it compare against momentum methods, such as the most widely used Adam(W)?

---

> ### Author Response · Authors · 2025-12-01
> **Thank you for your thoroughreview and enriching feedback. Below is our response**
>
> - **What metric is used.** The Euclidean geometry on $\\mathbb{R}^{d+1}$ is indeed adopted for Equation (3), which indeed means that RBO treats loss and parameters interchangeably. We have considered using a skewed metric (in the style of proximal optimization), but that seemed to destroy the physical intuition for our algorithm (it is much harder to reason about the motion of rigid bodies in non-Euclidean space). This is not to say that this extension (and the connection of RBO to the proximal point method and trust region methods) is not a future question we intend to work on. Another reason why we did not pursue this direction in the present work is the fact that we already treat parameter space as if it were homogeneous and isotropic, which is already unnatural.
> - **Scaling and symmetries.** To directly answer the reviewer's question, yes, multiplying loss by a constant changes RBO's behavior. This is however not a problem because:
>     1. Gradient descent isn't equivariant with respect to loss rescaling either, unless parameters are also rescaled, in which case RBO is equivariant as well. In fact, for RBO, this scenario can be elegantly interpreted as simply changing the radius.
>     2. The sensitivity to scaling is in fact the expected behavior, since scaling the loss up makes the valleys sharper, and hence, renders some minima inaccessible. We argue this is a feature, not a bug.
> - **Projection details.** 10 projection steps were used for all RBO instances with a step size $10^{-3}$. We opted for a specified number of steps rather than a stopping criterion. We initially aimed to experiment with different projection solvers (Particularly, Conjugate Gradient, and the use of preconditionners), but GD ended up producing satisfactory results for our experiments. We haven't used a schedule for any hyperparameter. Non-unique projections are not handled explicitly. In fact, we argue it is impossible to handle them (since the solution would involve integrating the downwards force over the possibly uncountable set of projections). We opted to ignore this issue, since for large radii, almost all the local minima of the distance are within a negligible distance of each other.
> - **Complexity analysis.** Counting the total number of forward and backward passes through the network, one epoch of RBO runs in $\\mathcal{O}(B\\times P)$, where $B$ is the number of batches, and $P$ is the number of projection iterations.
> - **Convergence analysis.** We are still working to establish convergence, complicated somewhat by the fact that offset manifolds are not $\\mathcal{C}^{\\infty}$-smooth (or even everywhere differentiable), even if the original function is. Currently, we are able to prove that RBO is equivalent to GD on the offset manifold, meaning that if it converges, it converges to a critical point of $f_\\rho$. Furthermore, we have proved that $f_{\\rho}$ is Lipschitz-smooth, meaning that all that is left to establish convergence is strong convexity (or a weaker condition like the PL-inequality).
> - **Empirical evaluations.**
>     - *Not enough epochs.* We are taking this into consideration, and increasing the number of epochs. However, we are not yet able to report full results on this yet. We note that the fact that the training curves of the different optimizers seem to be already flattening, it is unlikely that they will cross if given more time.
>     - *Baseline tuning.* Noted, and is being implemented. We note though that the exclusion of momentum was deliberate, because the effect of momentum is completely orthogonal to that of RBO, and they can be deployed together.
>     - *Missing ablations.* Tolerance and projection error are not investigated as the true projection is unknown, and therefore distance to it can't be estimated. As for the effect of the projection method on the final performance, we distinguish two cases: that of small radii, and that of large radii. In the former case, RBO reduces to SGD regardless of the behavior of the projection method. As for the latter, inaccuracy in the projection process is tolerable as long as it remains negligible compared to the radius.
>     - *Theory is partial.* We acknowledge that this is a limitation of our work in the conclusion, and are working towards addressing it.
> - **Exact form of $\\tau$ and $\\nu$.**  $\\tau = \\begin{bmatrix} \\nabla f(\\theta) \\\\ \\|\\nabla f (\\theta)\\|^2\\end{bmatrix}$, $\\nu = \\frac{1}{\\sqrt{1 + \\|\\nabla f(\\theta)\\|^2}} \\begin{bmatrix} \\nabla f(\\theta) \\\\ -1\\end{bmatrix}.$ These have been added to the paper
> - **Reduction to GD.** One proves easily that, as $\\rho \\to 0$, $f_{\\rho} \\to f$ locally uniformly. This is the rigorous version of the heuristic statement we made in the abstract.
> - **Comparison to momentum.** RBO is marginally outperformed by Adam in most scenarios we have experimented with.

---

### Official Review · Reviewer_VZaA · 2025-11-01

**Soundness:** 3
**Presentation:** 3
**Contribution:** 3
**Rating:** 4
**Confidence:** 3

**Summary:**

The paper introduces the Rolling Ball Optimizer, which simulates rolling a voluminous object down the loss landscape rather than a point mass. It does this using a projected gradient descent approach. The authors show that under some idealized conditions, this has a nice "ironing" property on the loss landscape, show empirical comparisons to methods like SAM, and investigate the role of the radius hyperparameter.

**Strengths:**

The idea, or especially its implementation, seem novel and yet intuitive.

The explanations for why it might work also seem to pass muster (learning rate and the radius phase transition).

**Weaknesses:**

Some of the motivation in the abstract and intro feels like overselling the problem, e.g., for a while it was believed that local minima might simply not exist in neural networks; see https://arxiv.org/pdf/1910.00359.

I would like to know _how_ much more computationally expensive this is; my intuition about doing the projections says "much more than SAM", which doesn't bode too well given they were trading blows in Table 1. In any case, that concern makes me want for some compute-matched experiments, to apply to settings where compute is the constraint rather than data, such as pretraining foundation models. Figure 4 gives the impression that RBO might be worse on such a basis unless it's only ~2x the cost of SAM.

The settings also seem impoverished, only going up to CIFAR-100 and ResNet-8s/VGG-9s, needlessly so, I suspect, if this could all be done with an RTX 4050.

**Questions:**

What's stopping the projection from creating another intersection with the landscape? Why is it safe to assume Equation (1) for the proofs?

---

> ### Author Response · Authors · 2025-12-01
> **Thank you for your review, we have attempted to address them**
>
> - **Some of the justifications are oversold.** The problematic geometries discussed in the abstract and introduction are indeed present in neural loss landscape (as confirmed by the paper the reviewer cites). We acknowledge that their impact on measured performance is not well-understood, and the community seems to have no consensus on the matter (as is demonstrated for example by the debate on the relationship between generalization and sharp minima).
> - **Computational complexity.** RBO's runtime is of order $\mathcal{O}(T\times P)$, where $T$ is the number of epochs and $P$ is the number of projection steps. SGD and its family of optimizers run in $\mathcal{O}(T)$ time. As for SAM, it also has runtime $\mathcal{O}(T\times P)$, where $P$ stands for one plus the number of iterations in the inner maximization loop. The original SAM paper set $P=2$, at the cost of limiting the validity of their update rule to the limit of $\rho\to 0$. It should be noted that other SAM implementations (like https://optax.readthedocs.io/en/latest/_collections/examples/contrib/sam.html) leave the choice of $P$ to the user, as we do. Our Experiments were conducted with $P=2$ for SAM (following the original implementation) and $P=10$ for RBO. Finally, we note that Entropy-SGD, The Newton-Raphson method, and Natural Gradient Descent all have the same order of complexity, despite performing no better (sometimes worse) than gradient-based optimizers. In the same vein, we believe RBO is interesting primarily as a tool to understand neural loss landscape, and secondarily as a practical learning algorithm (particularly in situations where robustness is required, which is the subject of a follow-up paper currently in writing).
> - **Compute matched experiments.** Preliminary results with this task demonstrate that RBO outperforms both SGD and SAM. It is only marginally outperformed by Adam.
> - **The settings also seem impoverished.** All the reported results were obtained using a Dell XPS 15 9530 equipped with an NVIDIA GeForce RTX 4050 GPU with 6141 MiB of VRAM running CUDA 13.0.
> - **Collisions during the projection.** It is indeed likely that the projection is not exact. However, one only needs the computed projection to be within a small distance (compared to the radius) of the true projection. A bit of heuristic geometric reasoning lends intuitive credence to this idea: the distance to the loss landscape increases in most directions, leaving a relatively small region where minima can be (unless the ball is stuck at the moth of a valley, which is indeed a failure case of RBO). As for Equation (1), it is exactly satisfied by an idealized version of RBO that exactly follows the motion of a ball instead of approximating it. We take this as an acceptable assumption for theoretical analysis, but make no use of it in our implementation or experiments.

---

### Meta-Review · Area_Chair_ix9r · 2026-01-07

**Summary:**

All reviewers voted to reject this paper. Their main concerns were with missing discussions (scale invariances, sensitivity to reparameterization) and algorithmic details, a lack of convergence guarantees, and a weak evaluation section. I am not convinced that the paper can overcome these concerns in this cycle. I vote to reject.

**Reviewer Concerns:**

Please see above.

**Reviewer Scores:**

I believe reviewers would have maintained or reduced their scores.

---

### Decision · Program_Chairs · 2026-01-26

Reject